

# Accounting for the vertical distribution of emissions in atmospheric CO$_2$ simulations

Dominik Brunner[1], Gerrit Kuhlmann[1], Julia Marshall[2], Valentin Clément[3,4], Oliver Fuhrer[4], Grégoire Broquet[5], Armin Löscher[6], and Yasjka Meijer[6]

[1]Empa, Swiss Federal Laboratories for Materials Science and Technology, Überlandstrasse 129, 8600 Dübendorf, Switzerland
[2]Max Planck Institute for Biogeochemistry (MPI-BGC), Jena, Germany
[3]Center for Climate Systems Modelling (C2SM), ETH Zurich, Zurich, Switzerland
[4]MeteoSwiss, Kloten, Switzerland
[5]Laboratoire des Sciences du Climat et de l'Environnement, CEA-CNRS-UVSQ, Université Paris Saclay, 91191, Gif-sur-Yvette cedex, France
[6]European Space Agency (ESA), ESTEC, Noordwijk, The Netherlands

**Correspondence:** Dominik Brunner (dominik.brunner@empa.ch)

**Abstract.** Inverse modeling of anthropogenic and biospheric CO$_2$ fluxes from ground-based and satellite observations critically depends on the accuracy of atmospheric transport simulations. Previous studies emphasized the impact of errors in simulated winds and vertical mixing in the planetary boundary layer, whereas the potential importance of releasing emissions not only at the surface but distributing them in the vertical was largely neglected. Accounting for elevated emissions may be critical,

since more than 50% of CO$_2$ in Europe is emitted by large point sources such as power plants and industrial facilities. In this study we conduct high-resolution atmospheric simulations of CO$_2$ with the mesoscale model COSMO-GHG over a domain covering the city of Berlin and several coal-fired power plants in eastern Germany, Poland and the Czech Republic. By including separate tracers for anthropogenic CO$_2$ emitted only at the surface or according to realistic, source-dependent profiles, we find that releasing CO$_2$ only at the surface overestimates near-surface CO$_2$ concentrations in the afternoon on average by 14% in

summer and 43% in winter over the selected model domain. Differences in column mean dry air mole fractions XCO$_2$ are smaller, between 5% in winter and 8% in summer, suggesting smaller yet non-negligible sensitivities for inversion modeling studies assimilating satellite rather than surface observations. The results suggests that the traditional approach of emitting CO$_2$ only at the surface is problematic and that a proper allocation of emissions in the vertical deserves as much attention as an accurate simulation of atmospheric transport.

*Copyright statement.* TEXT

## 1   Introduction

Reliably predicting future atmospheric concentrations of CO$_2$, the most important long-lived greenhouse gas, requires a profound understanding of the global carbon cycle, the contributions from anthropogenic and natural fluxes, and their sensitivity





to climate change and political and societal drivers. An important tool for advancing our knowledge of the carbon cycle is the integration of $CO_2$ observations with atmospheric transport simulations in an inverse modeling framework. Global inverse modeling systems helped to better constrain the terrestrial carbon budget, to allocate the global land sink to different continents and ecosystems, and to assess inter-annual variability and the sensitivity to climate variations (e.g. Bousquet et al., 2000;

Chevallier et al., 2010; Peylin et al., 2013; van der Laan-Luijkx et al., 2017; Rödenbeck et al., 2018). Mesoscale inverse modeling systems assimilating observations from dense, regional in situ networks are increasingly being used to study biospheric fluxes at the regional scale (e.g. Sarrat et al., 2009; Goeckede et al., 2010; Broquet et al., 2011; Meesters et al., 2012).

The Paris Climate Agreement adopted in 2015 (United Nations Framework Convention on Climate Change, 2016), which requires each signing partner nation to accurately report its GHG emissions and to reduce emissions in the future following

its Nationally Determined Contribution, has boosted the interest of the atmospheric science community in quantifying not only natural fluxes but also anthropogenic emissions of $CO_2$. "Top-down" inverse estimation of anthropogenic emissions from atmospheric observations has in fact been proposed as an independent method to complement the traditional "bottom-up" collection of national emission inventories (Nisbet and Weiss, 2010). Several initiatives at the national and international level such as the WMO's Integrated Global Greenhouse Gas Information System (IG3IS) have been launched to advance and

harmonize inverse methods with the long-term vision to establish these methods as a policy-relevant verification and support tool. A summary of the state of the science in inverse modeling for verification of greenhouse gas inventories has recently been presented by Bergamaschi et al. (2018a).

Because of the great challenge of accurately estimating $CO_2$ fluxes and distinguishing between the biospheric and anthropogenic contributions, the scientific community is calling for a globally integrated carbon observation and analysis system.

This system should build on a substantially expanded ground-based and satellite observation capacity and should integrate observations and bottom-up data with atmospheric transport modeling in an inverse framework (Ciais et al., 2014). The individual components and necessary steps towards an operational European system for quantifying anthropogenic $CO_2$ emissions were outlined in two recent reports to the European Commission (Ciais et al., 2015; Pinty et al., 2017).

A central component of this system is a constellation of $CO_2$ satellites with imaging capability similar to the CarbonSat

concept proposed by Bovensmann et al. (2010). These satellites should be able to quantify the $CO_2$ emissions from large sources such as power plants and cities during single overpasses. Corresponding observing system simulation experiments (OSSEs) were presented by Pillai et al. (2016) and Broquet et al. (2018).

In order to establish the requirements for such a constellation of satellites, the European Space Agency (ESA) has launched several scientific support studies including SMARTCARB, a project that focused on the potential of complementary satellite

measurements of $NO_2$ or CO to improve the quantification of $CO_2$ emissions from cities and power plants. The present study makes use of the high-resolution atmospheric transport simulations conducted in SMARTCARB to address a specific topic beyond of the main focus of the project. The core results of the SMARTCARB study will be presented elsewhere.

Estimating $CO_2$ fluxes by inverse modeling requires accurate simulation of observed atmospheric $CO_2$ concentrations. Systematic model biases are particularly critical as they tend to directly translate into biased flux estimates. Several studies

investigated the impact of uncertainties in atmospheric transport on simulated $CO_2$ concentrations and how they could be



accounted for in an inverse modeling framework (e.g. Lin and Gerbig, 2005; Chan et al., 2008; Lauvaux et al., 2009). A particular focus was placed on potential biases introduced by errors in vertical transport in the PBL (Gerbig et al., 2008; Kretschmer et al., 2012, 2014). Uncertainties associated with bottom-up $CO_2$ emissions were related to uncertainties in the horizontal gridding (Hogue et al.) or the representation of the temporal variability (Liu et al., 2017).

Uncertainties associated with the vertical placement of emissions, conversely, have been largely ignored so far. In fact, in the vast majority of $CO_2$ atmospheric transport and inverse modeling studies, $CO_2$ emissions were released exclusively at the surface (e.g. Sarrat et al., 2009; Lauvaux et al., 2009; Broquet et al., 2011; Ganshin et al., 2012; Meesters et al., 2012; Pillai et al., 2016; Lauvaux et al.; Liu et al., 2017; Graven et al., 2018; Fischer et al., 2018). This approach is questionable given the fact that a large proportion of $CO_2$ is released from point sources such as power plants well above the surface. Smoke stacks

are in fact designed to minimize the impact on concentrations at the ground. This is particularly important during situations with stable inversions in winter or on clear nights. However, even in well-mixed conditions, an elevated release may lead to a faster dilution and different propagation of the signal due to wind speeds and directions changing with altitude.

In the air quality modeling community, the importance of vertically distributing emissions is well established, especially for species such as $SO_2$ that are primarily emitted from power plants and industrial sources (Bieser et al., 2011; Mailler et al.,

2013; Karamchandani et al., 2014; Guevara et al., 2014). The main goal of this study is to demonstrate that this is also critical for the simulation of atmospheric $CO_2$ concentrations.

We employ very high resolution (1.1 km × 1.1 km) simulations for the year 2015 conducted over a 750 km x 650 km wide domain centered on the city of Berlin, Germany. The simulations included separate tracers representing $CO_2$ emitted only at the surface and $CO_2$ emitted according to source-dependent vertical profiles. As the domain covered several large coal-fired power

plants, we also investigate the impact of dynamically accounting for plume rise versus applying static vertical emission profiles. We focus on domain-averaged statistics rather than on individual plumes and on near-surface concentrations in the afternoon and total columns at satellite overpass time as would typically be used in inverse modeling. Conditions with a well-mixed and fully-developed boundary layer during daytime are expected to minimize the mismatch between model and observations.

## 2   Data and methods

### 2.1   COSMO-GHG model

COSMO is a limited-area, non-hydrostatic numerical weather prediction (NWP) model developed by the German weather service together with a consortium of seven European weather services (Baldauf et al., 2011). In addition to operational weather prediction, the model is applied widely for climate and air pollution research in various modified and extended versions (e.g Rockel et al., 2008; Davin et al., 2011; Vogel et al., 2009; Zubler et al., 2011), making COSMO a versatile community model.

In the project CarboCount-CH (Oney et al., 2015), COSMO was extended with a module for the simulation of greenhouse gases, hereafter referred to as COSMO-GHG. The extension was built on a newly developed tracer module (Roches and Fuhrer, 2012), which replaced the previous nonuniform treatment of meteorological tracers in the model. A similar, independently developed COSMO-based $CO_2$ model system was presented by Uebel and Bott (2018).



COSMO-GHG was first applied by Liu et al. (2017) to study the spatiotemporal patterns of fossil fuel $CO_2$ in Europe. They concluded that fossil-fuel $CO_2$ accounts for more than half of the total (anthropogenic plus biospheric) temporal variations in atmospheric $CO_2$ over large parts of Europe. Furthermore, they evaluated the model against observations demonstrating that the simulated fossil fuel variations favorably agreed with observations of fossil fuel $CO_2$ derived from concurrent measurements

of CO and $^{14}CO_2$.

COSMO is the first NWP model worldwide that is running operationally on hardware accelerated using graphical processing units (GPU) (Fuhrer et al., 2014). This highly efficient code is operationally used by the Swiss weather service MeteoSwiss and has been applied for decadal convection-resolving climate simulations (Leutwyler et al., 2017). In the framework of SMARTCARB, the modules of the GHG extension were ported to the GPU version following the concept of OpenACC

compiler directives, which is a high level approach to offload compute intensive part to a GPU accelerator (Lapillonne and Fuhrer, 2014). Benchmark tests showed that the GPU version achieved a speedup by a factor of six, which allows to increase the spatial and time resolution and greatly reduced the computational (and energy) costs of the simulations conducted in this study.

## 2.2 Model domain and setup

The model simulations were conducted in the framework of the project SMARTCARB, which aimed at studying the potential benefit of adding a $NO_2$ or CO channel to the instrument package of a future $CO_2$ satellite mission with respect to quantifying $CO_2$ emissions from strong localized sources such as cities and power plants. For this purpose, a model domain was selected covering the city of Berlin as well as several large power plants in Germany, Poland and the Czech Republic. The domain extended approximately 750 km in the east-west and 650 km in the south-north direction to cover at least a complete 250 km

wide satellite swath on either side of the city. Berlin was selected not only because it is one of the largest cities in Europe, but also because it is rather isolated, and because it has already been investigated in previous $CO_2$ modeling and observation studies (Pillai et al., 2016; Hase et al., 2015). Simulations were conducted for the complete year 2015 at 1.1 km x 1.1 km horizontal resolution and 60 vertical levels with a top at 24 km. The lowest model layer had a thickness of 20 m. The high resolution was selected to comply with the small pixel size of 2 km x 2 km of the planned $CO_2$ imaging satellite, keeping in

mind that the effective resolution of Eulerian transport models is much lower than the spacing of the model grid (Kent et al., 2014).

The simulations included a total of 50 different passively transported tracers of $CO_2$, CO and $NO_x$. The tracers represented different sources, release times or release altitudes and included background tracers constrained at the lateral boundaries by global-scale models. Two additional tracers for biospheric fluxes (respiration and photosynthesis) were included for $CO_2$, and

four additional tracers with varying e-folding lifetimes of 2, 4, 12 and 24 hours for $NO_x$. In this study, however, we focus on only a few of these tracers, notably on the following four anthropogenic $CO_2$ emission tracers:

- CO2_VERT: All anthropogenic emissions in the model domain released according to source-specific vertical profiles.

- CO2_SURF: Same as CO2_VERT but all emissions released at the surface, i.e. into the lowest model layer.



- CO2_PP-PR: Emissions from the six largest power plants with explicit plume-rise calculations.

- CO2_PP-EMEP: Emissions from the six largest power plants released according to a fixed vertical emission profile.

The simulations were nested into the operational European COSMO-7 analyses of MeteoSwiss, which provided the lateral boundary conditions for meteorological variables (temperature, pressure, wind, humidity, clouds) at 7 km horizontal and hourly temporal resolution. Because of the relatively small domain, these boundary conditions provided a strong constraint for the meteorology within the domain. For $CO_2$, lateral boundary conditions were obtained from a global free-running high-resolution $CO_2$ simulation (T1279, 137 levels, ~15 km horizontal resolution) provided by the European Center for Medium Range Weather Forecast (ECMWF) through the European Earth observation program Copernicus (Agustí-Panareda et al., 2014). The simulation was indirectly constrained by in-situ data by using a global climatology of optimized biospheric fluxes computed with the $CO_2$ assimilation system of Chevallier et al. (2010).

## 2.3 Emissions and biospheric fluxes

Anthropogenic emissions were obtained by merging the European TNO/MACC-3 inventory with a detailed inventory available for the city of Berlin. Dedicated municipal inventories can accurately account for the specific conditions and activities in a city and may, therefore, significantly deviate from inventories obtained by simple downscaling from larger scale inventories (Timmermans et al., 2013; Gately and Hutyra, 2017).

Version 3 of the TNO/MACC inventory was used, which has an improved representation of point sources as compared to version 2 described in Kuenen et al. (2014). It has a nominal resolution of 1/16° x 1/8° (approximately 7 km x 7 km) but additionally reports the emissions from strong point sources at their exact location. Emissions are provided separately for different source categories following the Standardized Nomenclature for Air Pollutants (SNAP) classification (EEA, 2002). The emissions of the year 2011 were taken, which was the most recent year available from the inventory.

The city of Berlin has developed a very detailed inventory for about 30 air pollutants and greenhouse gases for seven major source categories. The inventory was provided in a Geographic Information System (GIS) format as a collection of shape files representing individual area, point and line sources. The latest available year was 2012. Since the source categories did not follow the SNAP nomenclature, a mapping onto SNAP categories was applied to ensure consistency with the TNO/MACC-3 inventory.

Both inventories were projected onto the COSMO model grid (rotated latitude/longitude grid with north pole at 43°N and 10°W, 0.01° x 0.01° resolution) using mass-conservative methods. Point sources were placed into the proper COSMO grid cell. As a last step, the two inventories were merged using a mask for the city of Berlin.

Fig.1 presents a map of the $CO_2$ inventory of Berlin in the original format and after projection onto the COSMO grid. The rasterized map reveals 22 strong $CO_2$ point sources, which together account for 41% of the total emissions in the city. The two horizontal stripes correspond to the paths of airplanes taking off and landing at the two main airports.

In order to calculate hourly emissions as input for the model simulations, temporal scaling factors were applied describing diurnal, day-of-week and seasonal variations. The same temporal profiles were used as in Liu et al. (2017), which are based



on factors originally developed for air pollution modeling (Builtjes et al., 2003). These profiles depend on SNAP category and, except for diurnal profiles, on country. Diurnal profiles were matched to the local time of Germany.

Biospheric $CO_2$ fluxes due to gross photosynthetic production and respiration were computed at the resolution of the COSMO model using the Vegetation Photosynthesis and Respiration (VPRM) model (Mahadevan et al., 2008). This diag-
nostic model is based on meteorological input (2-m temperature and downward shortwave radiation at the surface) along with the Enhanced Vegetation Index (EVI) and Land Surface Water Index (LSWI) calculated from MODIS satellite reflectances. These indices were available as an 8-day product (MOD09A1, V006) at 500 m spatial resolution. Vegetation classes were determined from the 1-km SYNMAP land cover map (Jung et al., 2006). Further parameters required for VPRM were based on fits to flux tower measurements representative of these vegetation classes. The model was run off-line for the whole year
2015, driven by the highest resolution ECMWF meteorological data available.

## 2.4  Vertical allocation of emissions and plume rise

Emissions were distributed in the vertical using source-specific profiles developed for the European Monitoring and Evaluation Program (EMEP) (see e.g. Bieser et al. (2011)). For the purpose of the present study, the number of vertical layers was increased from seven to ten to enable a finer allocation to the model layers of COSMO-GHG. Specifically, the layers 4-90 m and 90-
170 m of the original EMEP profiles were divided into three and two sub-layers, respectively. The profile for SNAP 9 (waste) was modified in two ways: (i) by moving 10% from higher layers to the lowest layer to account for $CO_2$ emissions from landfills and waste water treatment plants, and (ii) by moving the large fractions originally placed into the layers 170-310 m (40%) and 310-470 m (35%) into lower layers between 90 m and 310 m. This modification was made following the study of Pregger and Friedrich (2009), which showed that the emission-weighted height of waste incinerator stacks in Germany is only
about 100 m and that 90% of the corresponding emissions including plume rise are expected to occur below 300 m.

The modified EMEP profiles are presented in Tab. 2 and illustrated in Fig. 2a. They were only applied to point sources such as power plants and large industrial facilities. A different set of profiles with lower average emission heights was applied to area sources, as these represent much weaker and more dispersed sources. The corresponding profiles are presented in Tab. 3 and Fig. 2b. The motivation for this distinction is best illustrated for SNAP category 2, residential and other non-industrial
heating. In the case of point sources, these are large heating facilities such as combined heat and power plants with tall stacks. In the case of area sources, in contrast, these are mostly private heating systems releasing $CO_2$ through chimneys at roof level. This is reflected in our profiles for SNAP 2 area sources being limited to the layers between 4 m and 60 m above surface.

As noted by Bieser et al. (2011), the EMEP profiles are based on very limited information originally collected for the city of Zagreb, Croatia. They therefore proposed a different set of profiles based on plume rise calculations for a large number of
point sources across Europe. Their study indicated that the vertical placement of emissions in the EMEP profiles is too high for combustion processes (SNAP 1, 2, 3, and 9) but too low for production processes (SNAP 4 and 5). For SNAP 1 (power plants), for example, they proposed a median release height of about 300 m, whereas the median in the EMEP profiles is about 400 m.

Although we did not apply the profiles of Bieser et al. (2011), our modification of SNAP 9 and the distinction between point and area sources effectively reduces the emission heights of $CO_2$ compared to the standard EMEP profiles. Furthermore, for



the 22 largest point sources in Berlin as well as the 6 largest power plants in the model domain in Germany (Jänschwalde, Lippendorf, Boxberg, Schwarze Pumpe) and Poland (Turów, Pątnów), the static EMEP SNAP 1 profiles were replaced by dynamic plume rise simulations for each hour of the year.

The effective emission height can be much higher than the geometric height of a stack because of the momentum and buoyancy of the flue gas. In general, plume rise depends on stack geometry (height and diameter), flue gas properties (temperature, humidity, exit velocity) and meteorological conditions (wind speed, atmospheric stability). Plume rise and the vertical extent of the plumes were calculated using the empirical equations recommended by the Association of German Engineers (VDI - Fachbereich Umweltmeteorologie, 1985), which are based on the original work of Briggs (1984). Hourly profiles of wind and temperature for the year 2015 were extracted from the COSMO-7 analyses of MeteoSwiss at the locations of the individual stacks. For power plants outside Berlin, typical stack and flue gas parameters were mainly taken from published statistics for Germany (Pregger and Friedrich, 2009), since these parameters are not publicly available (see Table 1). For stacks in Berlin, parameters were included in the emission inventory provided by the city of Berlin.

Two complicating factors were not considered in the present study. The European standards for large combustion plants ($http://data.europa.eu/eli/dec\_impl/2017/1442/oj$) require the flue gas to be cleaned for sulfur- and nitrogen-oxides. As a consequence of the chemical washing process, the temperature of the flue gas is reduced to a level where it can no longer be released via a classical smoke stack (Busch et al., 2002). In order to avoid re-heating, the flue gas is therefore often directed to the cooling tower and mixed into its moist buoyant air stream. This is true for the major German power plants in the domain, whereas the power plants Turów and Pątnów in Poland were, to the best of our knowledge, still equipped with smoke stacks in 2015. Plume rise computations are much more complex in the case of cooling towers due to latent heat release and the interaction with moisture in the ambient air (Schatzmann and Policastro, 1984). The additional release of latent heat may enhance plume rise by 20% to 100% compared to a dry plume (Hanna, 1972). Second, large power plants such as Jänschwalde are equipped with multiple cooling towers a short distance from each other. Their plumes tend to interact in a way that enhances plume rise, an effect that additionally depends on the alignment of the towers relative to the flow direction (Bornoff and Mokhtarzadeh-Dehghan, 2001). Neglecting these effects may thus underestimate true plume rise in our calculations.

## 3 Results

### 3.1 Plume rise at power plants

An example for the results of the plume rise simulations are presented for Jänschwalde, the largest power plant in the domain. Figure 3a shows the hourly evolution of the plume center heights during the year 2015. Plumes typically rise 100 m to 400 m above the top of the cooling tower, but occasionally much further when both winds and atmospheric stability are low. Plume rise varies strongly from day to day and shows a pronounced seasonal cycle. Plumes rise on average to 360 m above ground in summer but only to 250 m in winter. Due to the diurnal evolution of the planetary boundary layer (PBL), plume rise also





varies with the hour of the day, except during winter. In summer, the amplitude of this diurnal variability is about 150 m, with a broad minimum at night from 21 to 08 LT (19 to 06 UTC) and a peak in the early afternoon.

Figure 4 compares the histograms of plume rise at Jänschwalde in summer and winter to the standard EMEP SNAP 1 profile of Fig. 2. In agreement with Bieser et al. (2011), the computed profiles tend to place emissions significantly lower

compared to EMEP, even in summer. Median and mean effective emission heights in 2015 were 266 m and 310 m above ground, respectively. For the smaller power plant Lippendorf, these numbers were 187 m and 210 m. Simulated plume rise for these two power plants was thus at least 100 m lower compared to the EMEP profile. The large fraction of emissions placed above 500 m in the EMEP profile seems particularly unrealistic. Our plume rise calculations are more consistent with the profiles recommended by Bieser et al. (2011).

**3.2  Emission profiles over Berlin**

The main emission sources of $CO_2$ in Berlin are "traffic", "private households and public buildings", "industry", "trade and others", and the "transformation sector" which includes large facilities for heat and electricity production. In 2015, traffic only accounted for 29% of all emissions, households and industry for 27% (Amt für Statistik Berlin-Brandenburg, 2018). By far the largest single source was the transformation sector with a share of 42% of the total. As a result, a significant fraction of

emissions is released through stacks.

Figure 5 shows the monthly mean vertical profiles of $CO_2$ emissions from Berlin in a winter month (February) and a summer month (August) as used in our simulations. The profiles reflect the superposition of emissions from different sectors with different vertical profiles (see Fig. 2) and different seasonal contributions. In February, for example, residential heating is an important source between 4 m and 60 m above ground, whereas emissions in these layers are almost completely absent

in August. Although in both months the profiles have a pronounced peak at the surface, 36% of $CO_2$ is released above 90 m in February, with that share rising to 58% in August when residential heating is small. These numbers suggest that a proper vertical allocation of emissions is important even in cities.

The vertical profiles of the emissions of CO and $NO_x$ are overlaid in Fig. 5 for comparison, since coincident measurements of $NO_2$ or CO may be used by a future $CO_2$ satellite for emission quantification. Both CO and $NO_x$ have a more pronounced peak

at the surface due to the larger proportion of traffic emissions. Similar to $CO_2$, a large (albeit smaller) fraction of $NO_x$ is emitted well above ground by large point sources. Interestingly, these sources emit very little CO, suggesting efficient combustion or cleaning of the exhaust. The fraction of CO released above 90 m is below 5%, in sharp contrast to $CO_2$.

**3.3  $CO_2$ at the surface**

Maps of the monthly mean afternoon $CO_2$ dry air mole fractions (hereafter referred to as "concentrations") in the lowest

model layer (0–20 m) from all anthropogenic emissions within the model domain are presented in Fig. 6 for January and July, respectively. Afternoon values were selected since current inversion systems usually assimilate only observations in the afternoon when vertical concentration gradients are smallest (e.g. Peters et al., 2010). The left hand panels show the results for the tracer CO2_VERT, with vertically distributed emissions, while the right hand panels show the tracer CO2_SURF, with all





emissions concentrated at the surface. For all power plants labeled in the figure, plume rise was explicitly simulated for the tracer CO2_VERT.

The concentrations are generally higher in January than in July due to higher emissions and reduced vertical mixing in winter. The concentrations are also significantly higher when all $CO_2$ is released at the surface. The main reason for these
differences are power plants and other point sources, which stand out prominently in the maps for the tracer CO2_SURF but not for CO2_VERT.

As shown in Fig.7, the differences are much larger in winter than in summer. In summer, power plant emissions are mixed efficiently over the depth of the afternoon PBL. Since this mixing is not instantaneous, differences are noticeable close to the sources but fade out rapidly with increasing distance. In winter, conversely, when vertical mixing is weak, the differences
between the two tracers remain well above 1 ppm over distances of a few tens of kilometers downstream, occasionally over 100 km or more.

Domain-averaged monthly mean diurnal cycles of the two tracers are presented in Fig. 8 for the months of January and July. Consistent with the maps, the concentrations of CO2_SURF are significantly higher than those of CO2_VERT. This difference is on the order of 1.6 ppm in January and almost constant over the day. The variations over the day appear to be dominated by
the diurnal cycle of the emissions rather than by the dynamics of the PBL.

In summer, the differences are generally smaller and tend to be very small in the afternoon, consistent with Fig. 7b. Due to the low PBL at night, the concentrations increase over the night despite relatively low emissions. This increase is much more pronounced for tracer CO2_SURF, which is susceptible to surface emissions from point sources that do not stop at night. The tracer CO2_VERT only shows a marked increase during the early morning hours when traffic increases and the PBL is still low.
Emissions from point sources, on the other hand, are likely released above the nocturnal PBL leading to marked differences between CO2_VERT and CO2_SURF at night.

Statistics of the afternoon concentrations of the two $CO_2$ tracers are summarized in Tab. 4 in terms of mean values and different percentiles of the frequency distribution. Domain-averaged $CO_2$ concentrations are 43% higher in January when all $CO_2$ is released at the surface compared to when emissions are distributed vertically. The differences are larger for the high
percentiles, suggesting that background values are less affected than peak values. This is understandable as vertical mixing tends to reduce the differences with increasing distance from the sources.

In summer, the differences are generally much smaller, but as suggested by Fig.8, this is only true for afternoon concentrations. Mean differences in the afternoon are of the order of 14%. Again, higher percentiles tend to show larger differences.

### 3.4 Column mean dry air mole fractions XCO$_2$

As shown in the previous section, the choice of vertical allocation of the emissions has a large impact on ground-level concentrations. Since the tracers CO2_VERT and CO2_SURF are based on the same mass of $CO_2$ emitted into the atmosphere and only differ by the vertical placement of this mass, differences in column mean dry air mole fractions (XCO$_2$) are expected to be small. However, since wind speeds tend to increase with altitude, $CO_2$ emitted at higher levels is more likely to be transported away from the sources and leave the model domain more rapidly.





Maps of $XCO_2$ and of the differences between the two tracers are presented in Figures 9 and 10 in the same way as for the surface concentrations. Instead of presenting mean afternoon values, the figures show the situation at 11:30 LT (the average of output at 10 UTC and 11 UTC) corresponding to the expected overpass time of the planned European satellites. Note that we did not account for daylight savings time in the diurnal cycles of emissions but assumed a constant offset of +1 hour

between local time in the domain and UTC. Differences in $XCO_2$ between the two tracers are indeed much smaller than the differences in surface concentrations, suggesting that total column observations are much less sensitive to the vertical placement of emissions. However, differences are not negligible, especially in winter (Fig. 10). The largest differences in January are seen in the northern parts of the Czech Republic, which is the main coal-mining region of the country featuring a large number of power plants. Since plume rise was not explicitly calculated, CO2_VERT emissions from these power plants followed the

EMEP profiles, which tend to place emissions too high, as mentioned earlier. Differences over the power plants with explicit plume rise are smaller, but not negligible, with values on the order of 1 ppm close to the sources.

Domain-averaged mean diurnal cycles are presented in Fig. 11 and overall statistics in Table 4. Similar to the results for near-surface concentrations, the differences are larger in winter than in summer. In contrast to the situation at the surface, the differences in $XCO_2$ remain fairly constant over the day not only in winter but also in summer. Relative differences in

mean $XCO_2$ are only 7.7% in January (compared to 43% at the surface) and 4.8% in July (compared to 14% at the surface). Note that the differences in the columns are related to the synthetic nature of our model experiment, since no anthropogenic $CO_2$ emissions are advecting into the domain from sources outside. Nevertheless, the analysis reveals significant differences in the contributions from sources within the domain, which is the information used by any regional inverse modelling system. Consistent with the findings for near-surface concentrations, the differences tend to increase for higher percentiles, reaching

about 10% at the 95th percentile.

Although differences in monthly mean $XCO_2$ values are relatively small, the differences can be very large at a given location and time as illustrated in Fig. 12 for 2 July 2015. The figure shows the differences in $XCO_2$ between two $CO_2$ tracers representing emissions from the largest power plants in the domain. The two tracers were released either using explicit plume rise simulations (CO2_PP-PR) or according to EMEP SNAP-1 profiles (CO2_PP-EMEP). No power-plant-only tracer was

simulated with emissions at the surface. Plume rise was rather moderate (to about 340 m above ground) at this time due to pronounced easterly winds. The red (positive) parts in the figure correspond to plumes produced by CO2_PP-PR whereas the blue parts (negative) correspond to the tracer CO2_PP-EMEP.

Due to wind directions changing with altitude and due to the different emission heights of the tracers, the plumes are transported in different directions. Spiraling wind directions are typical of the boundary layer where winds near the surface

have an ageostrophic, cross-isobaric component due to surface friction, while winds become increasingly geostrophic at higher levels. This is known as the Ekman spiral. The plumes of CO2_PP-EMEP show a stronger lateral dispersion because the tracer is released over a large vertical extent (blue line in Fig. 2). The vertical cross-section transecting the plumes about 15 km to 30 km downwind of the sources suggests that both tracers are partially mixed over the full depth of the boundary layer at this distance, but that the centers of the plumes are clearly higher above the surface for the tracer CO2_PP-EMEP than for

CO2_PP-PR.





## 4 Discussion

The results for near surface concentrations revealed a strong sensitivity to the vertical placement of emissions, especially in winter. Similarly large sensitivities were reported for air pollution simulations. By conducting a set of five one-year European scale model simulations differing only in the vertical allocation of emissions, Mailler et al. (2013) found that ground-based concentrations of $SO_2$, an air pollutant primarily released by point sources, increased on average by about 70% when reducing all emission heights by a factor four. The changes were less significant (about 15%) for $NO_2$ due to the larger contribution from traffic emissions. Sensitivities of a similar magnitude were reported by Guevara et al. (2014) for simulations over Spain when replacing the EMEP profiles for power plants by more realistic plume rise calculations.

The results of the present study may be considered as upper limits of the sensitivity for two reasons. First of all, our simulations covered a region in Europe with a particularly high density of coal-fired power plants. Simulations over other regions such as France, where electricity is mostly produced by nuclear power, would likely have yielded lower sensitivities. Nevertheless, averaged over Europe, large point sources are responsible for at least half of all anthropogenic $CO_2$ emissions (52% in 2011) according to the TNO/MACC-3 inventory, which underscores the general importance of properly allocating their emissions vertically.

Second, for most point sources in our domain the standard EMEP profiles were applied, which tend to place emissions too high in the atmosphere as suggested consistently by Bieser et al. (2011); Guevara et al. (2014); Mailler et al. (2013) and by our own comparison with explicit plume rise simulations. Several improvements were already implemented in the present study, each of them leading to a reduction of the effective emission heights and likely to a more realistic representation as compared to EMEP. This included a modification of SNAP 9 profiles, the distinction between point and area sources, and the explicit computation of plume rise for the largest sources in the domain. Due to a lack of representative studies, these modifications were somewhat arbitrary. More studies like Bieser et al. (2011) and Pregger and Friedrich (2009) are needed, but should not only target large point sources but also emissions from the remaining 48% of emissions, including residential heating, even though their vertical placement will be less critical. Explicit plume rise computations for all large point sources as performed in some air quality models (Karamchandani et al., 2014) would be the best alternative to using static profiles, but this adds significant complexity to the model, and individual stack and flue gas parameters are not publicly available in Europe (Pregger and Friedrich, 2009).

None of the studies mentioned above considered the issue of interaction between multiple plumes and latent heat release in cooling tower plumes, which tend to enhance plume rise. The SNAP 1 profiles recommended by Bieser et al. (2011) and the simulations conducted here are only representative for isolated stacks, which is not consistent with any of the German coal-fired power plants in our domain. Comprehensive models for plume rise from single and multiple interacting cooling tower plumes have been presented by Schatzmann and Policastro (1984) and Policastro et al. (1994), but they seem not to be widely applied, although the code of Schatzmann and Policastro (1984) is available through the Association of German Engineers (VDI, $https://www.vdi.de/index.php?id = 4791$). Most plume rise studies date back 20 to 50 years, and modern computational




fluid dynamics simulations are lacking, with the exception of the study of Bornoff and Mokhtarzadeh-Dehghan (2001) on the interaction of plumes from two adjacent cooling towers.

Mean afternoon differences at the surface in January between the tracers CO2_VERT and CO2_SURF are of the order of 1.5 ppm, which is close to 50% of the total anthropogenic $CO_2$ signal due to emissions within the model domain. In summer, the relative differences in the afternoon are much smaller due to more efficient vertical mixing, but still on the order of 14%. Inaccurate vertical placement of the emissions may thus lead to biases of a magnitude comparable to other error sources reported in the literature. Random uncertainties on the order of 30-50% of the regional biospheric $CO_2$ signal have been estimated, for example, for model errors in PBL mixing (Gerbig et al., 2008) and in wind speed and direction (Lin and Gerbig, 2005). Systematic biases in simulated near-surface wind speeds of mesoscale weather prediction models like COSMO or WRF are typically on the order of 0.5-1 m s$^{-1}$ or about 10-20% of the mean wind speeds (Jiménez and Dudhia, 2012; Brunner et al., 2015; Bagley et al.), which may translate into similar biases in near-surface concentrations. Systematic differences in emission estimates using different regional transport and inverse modeling systems were reported to be on the order of 20%-40% (Hu et al.; Bergamaschi et al., 2018b). Our analysis focused on the situation in the lowest model layer at 0-20 m. However, the recommended strategy for $CO_2$ monitoring is to sample from tall towers well above the surface. At higher elevations the model sensitivity to the vertical placement of emissions is likely smaller, which is another benefit of tall tower measurements in addition to their greater spatial representativeness.

Mean differences in total column XCO$_2$ between the tracers CO2_VERT and CO2_SURF are much smaller, around 8% in winter and 5% in summer. However, differences are larger at the highest percentiles (about 10% at the 95% percentile), which are more relevant for satellite missions like CarbonSat designed to image individual plumes. Furthermore, the vertical placement of emissions has a significant effect on the speed and direction of individual plumes, suggesting that an accurate vertical placement is a critical requirement for inverse modeling of power plant emissions from satellite observations. Irrespective of a correct vertical placement of emissions, an appropriate simulation of power plant plumes will remain a great challenge for any mesoscale atmospheric transport model. Current approaches for estimating power plant emissions are circumventing this problem by directly matching the "observed" wind direction defined by the location of the plume, e.g., by fitting a Gaussian plume model (Krings et al., 2018; Nassar et al., 2017). However, also these methods need to make a realistic assumption about the height of the plume, since the mean advection speed of the plume needs to be estimated from a simulated or observed vertical wind profile.

## 5 Conclusions

We investigated the sensitivity of model-simulated near-surface and total column $CO_2$ concentrations to a realistic vertical allocation of anthropogenic emissions as opposed to the traditional approach of emitting $CO_2$ only at the surface. The study was conducted using kilometer-scale atmospheric transport simulations for the year 2015 for a domain covering the city of Berlin and numerous power plants in Germany, Poland and the Czech Republic. More than 50% of $CO_2$ in Europe is emitted from large point sources, mostly through stacks and cooling towers, suggesting that a proper representation of these sources in




the vertical dimension may be critical. Our results indeed confirm a strong sensitivity of near-surface afternoon concentrations: A regional $CO_2$ tracer released in the model domain only at the surface was on average 43% higher in winter and 14% higher in summer than a tracer released according to realistic vertical profiles. Differences were smaller but not negligible (5%-8%) for total column $XCO_2$, suggesting that the assimilation of satellite observations is less sensitive to the vertical placement of

emissions than the assimilation of ground-based observations. Individual plumes as imaged by future $CO_2$ satellites, however, may propagate more rapidly and in different directions when using realistic vertical profiles instead of releasing $CO_2$ only at the surface.

Plume rise was explicitly simulated for the six largest power plants in the domain and for the 22 largest point sources in Berlin. Power plant plumes typically rose between 100 m and 400 m above the top of the stacks. Plume rise showed significant

seasonal and diurnal variability that would be missed when applying static vertical profiles. Simulated plume rise was on average more than 100 m lower than suggested by the frequently used EMEP profile for power plant emissions. An accurate vertical placement of emissions is not only critical for power plants but may also be relevant for the simulation of city plumes. In the case of Berlin, for example, more than 35% of $CO_2$ is released through stacks presumably more than 90 m above surface.

We strongly recommend the representation of $CO_2$ emissions in all three dimensions in regional atmospheric transport and

inverse modeling studies. The specific impact on the model results will depend on factors such as vertical model resolution and boundary layer scheme, but in general cannot be expected to be negligible. Current gridded emission inventories only provide information in two dimensions. Information on the source-specific vertical allocation of emissions as used in this study, conversely, is still sparse and should receive more attention in the future.

*Data availability.* Column averaged dry air mole fractions of all simulated tracers are available both as 2D fields and as synthetic Level 2

satellite products through ESA. The total 3-dimensional model output amounts to 30 TB and is archived at Empa servers. Selected fields or time periods can be made available upon request. The TNO/MACC-3 inventory is available through Copernicus (http://macc.copernicus-atmosphere.eu). The emissions inventory of Berlin was kindly provided by Andreas Kerschbaumer, Senatsverwaltung Berlin, and is available for research upon request. The global $CO_2$ simulation, which provided the lateral boundary conditions, was conducted in the framework of Copernicus Atmosphere Monitoring Service and can be retrieved from ECMWF's archive as experiment gf39, stream lwda, class rd.

*Competing interests.* The authors declare that they have no conflict of interest.

*Acknowledgements.* This study was conducted in the context of the project SMARTCARB funded by the European Space Agency (ESA) under contract no. 4000119599/16/NL/FF/mg. The views expressed here can in no way be taken to reflect the official opinion of ESA. The work was supported by a grant from the Swiss National Supercomputing Centre (CSCS) under project ID d73. We would like to acknowledge Richard Engelen and Anna Agusti-Panareda (ECMWF) for providing support and access to global $CO_2$ model simulation fields which were

generated using the Copernicus Atmosphere Monitoring Service Information [2017].The TNO/MACC-3 emissions inventory and temporal





emission profiles were kindly provided by Hugo Denier van der Gon (TNO, The Netherlands). Finally, we are very grateful to Andreas Kerschbaumer, Senatsverwaltung Berlin, for providing the emission inventory of Berlin and additional material and for being available for discussions on its proper usage.



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





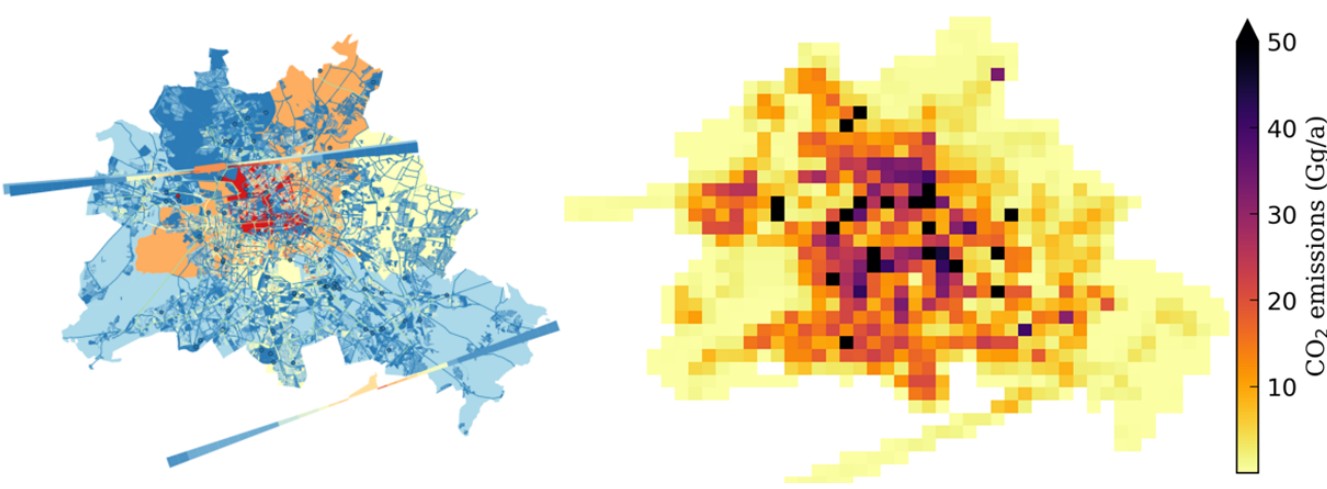

**Figure 1.** (Left) Sketch of $CO_2$ emission in the Berlin inventory with point, line and area sources from different emission categories. (Right) Total $CO_2$ emissions re-projected and rasterized onto the COSMO model grid.





**Figure 2.** Vertical emission profiles applied for (a) point sources and (b) area/line sources. The alternating gray and white backgrounds denote the vertical layers.



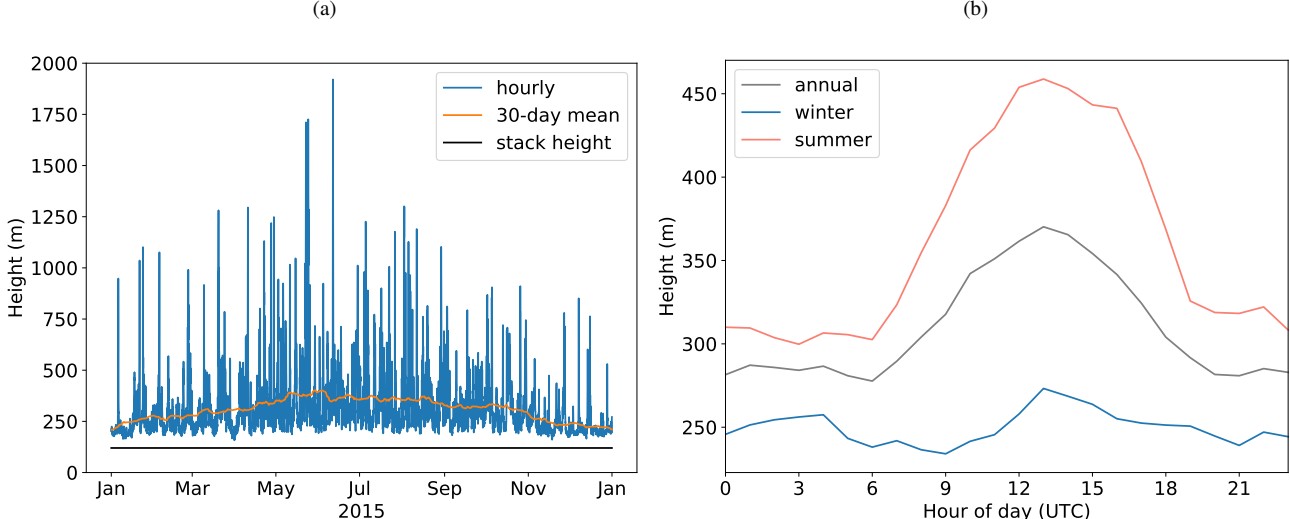

**Figure 3.** Effective $CO_2$ emission heights at the power plant Jänschwalde, Germany, based on plume rise calculations. (a) Hourly plume rise in 2015. The orange line is a 30-days moving average. The black solid line denotes the height of the cooling tower (120 m). (b) Mean annual, winter (DJF) and summer (JJA) diurnal cycles.

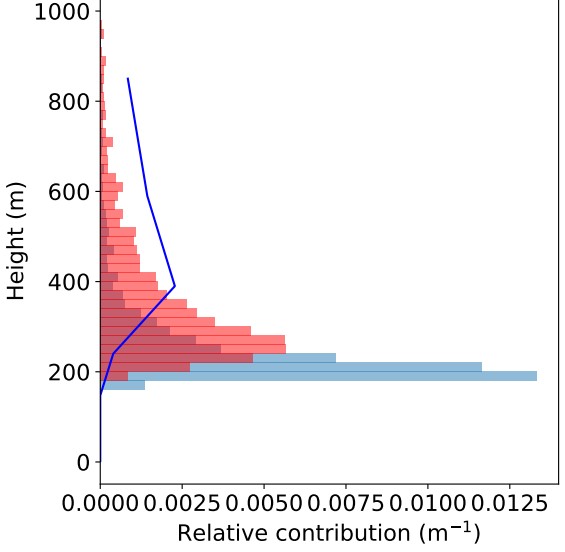

**Figure 4.** Normalized histogram of plume rise at the power plant Jänschwalde in winter (blue) and summer (red). The blue line shows the EMEP standard profile for power plants for comparison. Areas below the curves integrated along the vertical axis are normalized to 1.





**Figure 5.** Monthly mean vertical emission profiles of $CO_2$, CO and $NO_x$ over Berlin in (a) February and (b) August.







**Figure 6.** Mean afternoon (14-16 LT) near-surface $CO_2$ in (top) January and (bottom) July contributed by all anthropogenic sources in the domain. (Left) Tracer CO2_VERT released according to realistic vertical profiles. (Right) Tracer CO2_SURF released only at the surface.



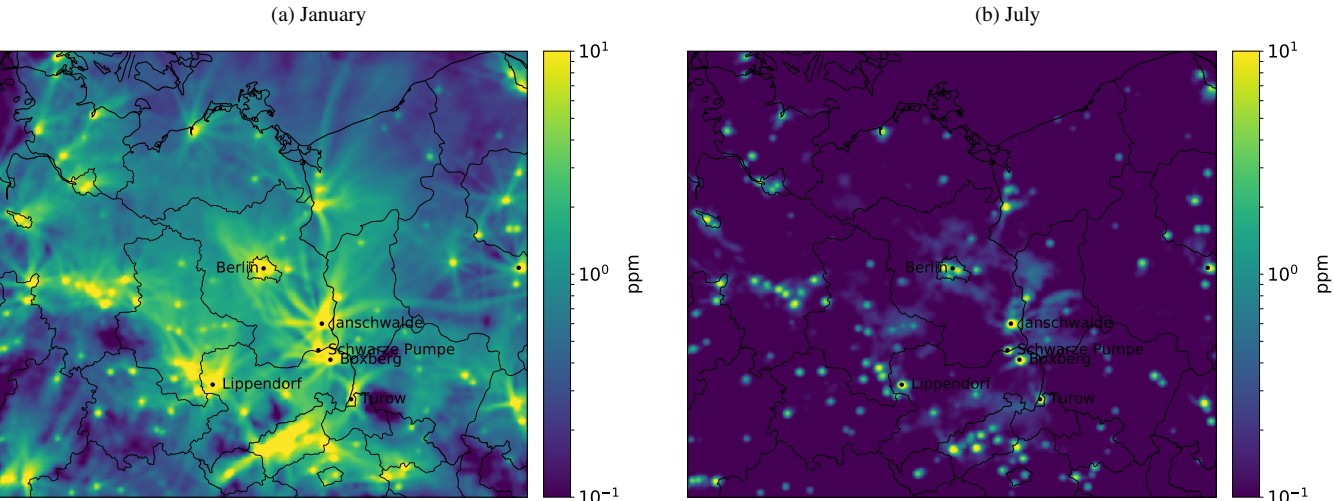

**Figure 7.** Difference in mean afternoon (14-16 LT) near-surface $CO_2$ between tracers CO2_SURF and CO2_VERT in (a) January and (b) July.

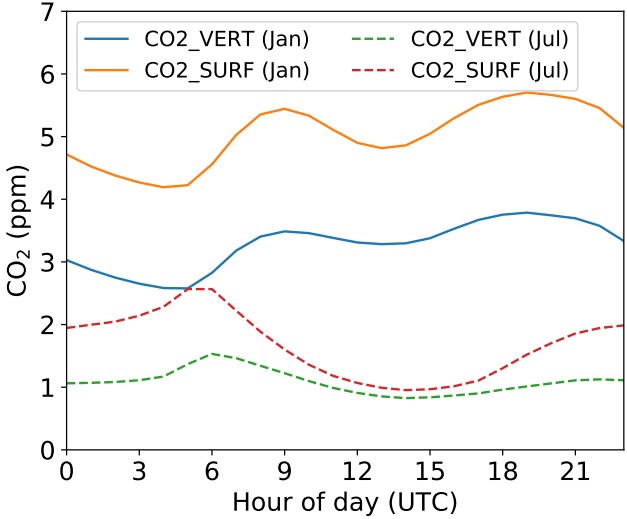

**Figure 8.** Mean diurnal cycles of the tracers CO2_VERT and CO2_SURF in January and July.





(a) CO2_VERT in January

(b) CO2_SURF in January

(c) CO2_VERT in July

(d) CO2_SURF in July

**Figure 9.** Mean 11:30 LT column mean dry air mole fractions (XCO$_2$) in (top) January and (bottom) July contributed by all anthropogenic sources in the domain. (Left) Tracer CO2_VERT released according to realistic vertical profiles. (Right) Tracer CO2_SURF released only at the surface.



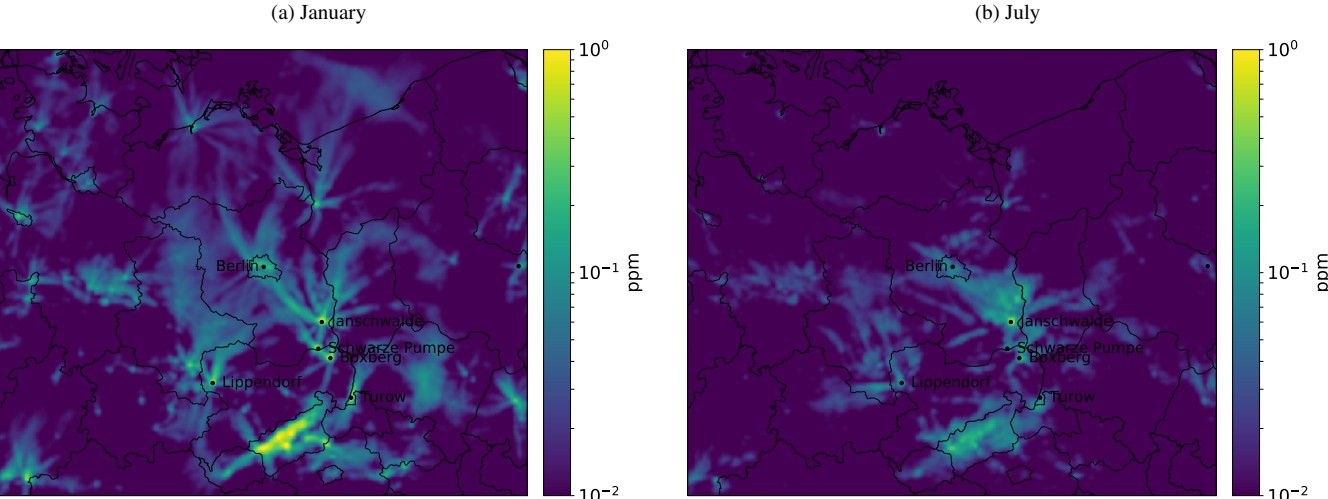

**Figure 10.** Difference in 11:30 LT column mean dry air mole fractions ($XCO_2$) between tracers CO2_SURF and CO2_VERT in (a) January and (b) July.

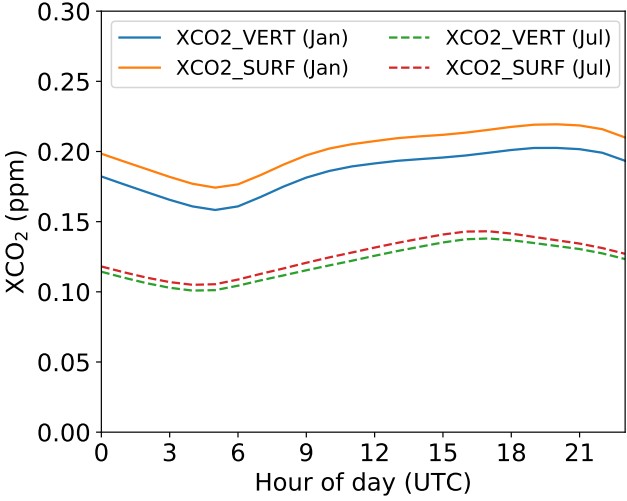

**Figure 11.** Mean diurnal cycles of column mean dry air mole fractions ($XCO_2$) of the tracers CO2_VERT and CO2_SURF in January and July.



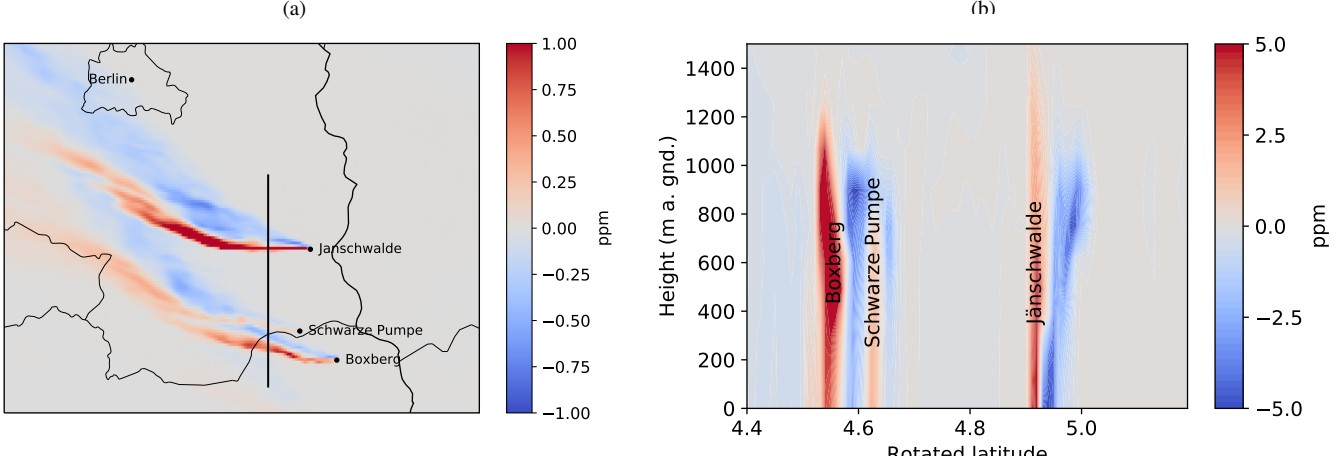

**Figure 12.** Difference on 2 July 2015 11:00 LT between power plant $CO_2$ tracer with explicit plume rise (CO2_PP-PR) and released according to standard EMEP SNAP-1 profile (CO2_PP-EMEP). (a) Map of the difference in column mean dry air mole fractions $XCO_2$. (b) Vertical cross-section of the difference in $CO_2$ along the transect indicated in (a). The length of this cross-section is about 100 km.

**Table 1.** Stack parameters used for plume rise calculation for six largest power plants in the model domain.

| Name | Longitude (° E) | Latitude (° N) | Stack height (m) | Effluent temperature[1] (K) | Volume flux[1] (m³ s⁻¹) |
|---|---|---|---|---|---|
| Jänschwalde | 14.458 | 51.837 | 120[2] | 322 | 790 |
| Lippendorf | 12.372 | 51.184 | 175[2] | 322 | 790 |
| Schwarze Pumpe | 14.354 | 51.538 | 141[2] | 322 | 790 |
| Boxberg | 14.569 | 51.418 | 155[2] | 322 | 790 |
| Turów | 14.911 | 50.948 | 150[1] | 416 | 159 |
| Pątnów | 18.238 | 52.303 | 114[1] | 447 | 59 |

[1] average parameters by fuel and plant capacity taken from Pregger and Friedrich (2009)

[2] https://de.wikipedia.org/wiki/Kühlturm (last access: 28 Aug 2018)





**Table 2.** Vertical emission profiles for point sources for different SNAP categories as fraction of the total emitted mass per vertical layer. Layers are denoted by their lower and upper limits (m above ground).

| SNAP | Description | 0–4 | 4–30 | 30–60 | 60–90 | 90–125 | 125–170 | 170–310 | 310–470 | 470–710 | 710–990 |
|---|---|---|---|---|---|---|---|---|---|---|---|
| 1 | Energy industry | 0.00 | 0.00 | 0.00 | 0.00 | 0.00 | 0.00 | 0.08 | 0.46 | 0.29 | 0.17 |
| 2 | Non-industrial combustion | 0.05 | 0.25 | 0.30 | 0.20 | 0.15 | 0.05 | 0.00 | 0.00 | 0.00 | 0.00 |
| 3 | Combustion in manufacturing industry | 0.00 | 0.04 | 0.08 | 0.12 | 0.16 | 0.20 | 0.20 | 0.15 | 0.05 | 0.00 |
| 4 | Production processes | 0.00 | 0.20 | 0.30 | 0.30 | 0.15 | 0.05 | 0.00 | 0.00 | 0.00 | 0.00 |
| 5 | Extraction/distribution of fossil fuels | 0.00 | 0.20 | 0.30 | 0.30 | 0.15 | 0.05 | 0.00 | 0.00 | 0.00 | 0.00 |
| 6 | Product use | 0.50 | 0.50 | 0.00 | 0.00 | 0.00 | 0.00 | 0.00 | 0.00 | 0.00 | 0.00 |
| 7 | Road transport | 1.00 | 0.00 | 0.00 | 0.00 | 0.00 | 0.00 | 0.00 | 0.00 | 0.00 | 0.00 |
| 8 | Non-road transport | 1.00 | 0.00 | 0.00 | 0.00 | 0.00 | 0.00 | 0.00 | 0.00 | 0.00 | 0.00 |
| 9 | Waste treatment | 0.10 | 0.00 | 0.10 | 0.15 | 0.20 | 0.20 | 0.15 | 0.07 | 0.03 | 0.00 |
| 10 | Agriculture | 1.00 | 0.00 | 0.00 | 0.00 | 0.00 | 0.00 | 0.00 | 0.00 | 0.00 | 0.00 |

**Table 3.** Vertical emission profiles for area sources for different SNAP categories as fraction of the total emitted mass per vertical layer. Layers are denoted by their lower and upper limits (m above ground). Emissions in SNAP 1 are released exclusively from point sources.

| SNAP | Description | 0–4 | 4–30 | 30–60 | 60–90 | 90–125 | 125–170 | 170–310 | 310–470 | 470–710 | 710–990 |
|---|---|---|---|---|---|---|---|---|---|---|---|
| 1 | Energy industry | - | - | - | - | - | - | - | - | - | - |
| 2 | Non-industrial combustion | 0.00 | 0.75 | 0.25 | 0.00 | 0.00 | 0.00 | 0.00 | 0.00 | 0.00 | 0.00 |
| 3 | Combustion in manufacturing industry | 0.00 | 0.25 | 0.25 | 0.25 | 0.25 | 0.00 | 0.00 | 0.00 | 0.00 | 0.00 |
| 4 | Production processes | 0.00 | 0.25 | 0.25 | 0.25 | 0.25 | 0.00 | 0.00 | 0.00 | 0.00 | 0.00 |
| 9 | Waste treatment | 0.20 | 0.10 | 0.25 | 0.25 | 0.20 | 0.00 | 0.00 | 0.00 | 0.00 | 0.00 |
| 5-8,10 | Other categories | 1.00 | 0.00 | 0.00 | 0.00 | 0.00 | 0.00 | 0.00 | 0.00 | 0.00 | 0.00 |





**Table 4.** Statistics of near surface $CO_2$ and column averaged dry air mole fractions $XCO_2$ in the model domain for the tracers CO2_VERT and CO2_SURF. Differences are presented in terms of (CO2_SURF-CO2_VERT)/CO2_VERT.

| Tracer | Month | Hour | Mean (ppm) | Median (ppm) | 25% (ppm) | 75% (ppm) | 95% (ppm) |
|---|---|---|---|---|---|---|---|
| CO2_VERT | January | 14-16 LT | 3.34 | 3.13 | 2.35 | 4.00 | 5.76 |
| CO2_SURF | January | 14-16 LT | 4.78 | 3.91 | 2.76 | 5.36 | 9.40 |
| Difference | | | 43% | 25% | 17% | 34% | 63% |
| CO2_VERT | July | 14-16 LT | 0.84 | 0.78 | 0.51 | 1.02 | 1.61 |
| CO2_SURF | July | 14-16 LT | 0.96 | 0.80 | 0.54 | 1.09 | 1.81 |
| Difference | | | 14% | 3% | 6% | 7% | 12% |
| XCO2_VERT | January | 11:30 LT | 0.193 | 0.185 | 0.145 | 0.230 | 0.317 |
| XCO2_SURF | January | 11:30 LT | 0.208 | 0.194 | 0.149 | 0.243 | 0.355 |
| Difference | | | 7.7% | 4.9% | 2.8% | 5.7% | 12.0% |
| XCO2_VERT | July | 11:30 LT | 0.125 | 0.110 | 0.067 | 0.170 | 0.253 |
| XCO2_SURF | July | 11:30 LT | 0.131 | 0.112 | 0.068 | 0.173 | 0.276 |
| Difference | | | 4.8% | 1.8% | 1.5% | 1.8% | 9.1% |