# Peer review of "Accounting for the vertical distribution of emissions in atmospheric CO2 simulations"

_Atmospheric Chemistry and Physics, 2018_

## Referee Comment (RC1) · Anonymous Referee #2 · 30 Dec 2018

General comments:

The proposed study investigates the impact on surface-level CO2 concentrations for simulations where all emissions are presumed to come from the surface versus from a more realistic vertical distribution. The authors establish that a vertical distribution of emissions is accepted as an important characteristic in the air quality community during their simulations of other, non-CO2 species, though it has been mostly neglected in the CO2 community. This is an important observation, and the authors even point out that large regional CO2 emitters such as power plants were specifically designed to emit at a higher altitude for the purpose of minimizing the impact on surface CO2 concentrations. They thus perform simulations for each emission type–surface and vertically distributed–to compare, and they further include a simulation testing the impact

of including plume dynamics. The hypothesis being tested looks to be of high value to the CO2 modeling community. The analysis used simulations from the SMARTCARB project and compared CO2 concentrations at times when uncertainty should be at its lowest, following standard practices from the modeling community. The plume analysis agrees with an earlier study's conclusion that EMEP vertical distributions may put emissions too high for combustion processes (i.e. powerplant stacks). The authors find that, compared to the standard surface-released CO2 emission scenario, one with a realistic vertical profile has decreased dry air mole fraction values at the surface layer in the afternoon (as is commonly used in inversion analyses), especially in the winter, where their January analysis showed a 43% mean difference (25% median difference). This is a significant finding that should be shared with the community. The authors further investigate the impact on total column XCO2 concentrations and find only a small difference in the monthly means, however a potentially significant difference near individual powerplant plumes. The difference in concentration values between power plant emissions simulated at-height versus with a plume-rise model are explored with individual tracers and found to be noteworthy because of the differing atmospheric dynamics. Their literature survey is well-performed and up-to-date, and their science is impactful and presented in a clear and straightforward manner. I recommend this manuscript for publication following minor edits.

Specific comments:

Page 4, Line 23: It might be good to include, after the sentence stating the height of the lowest vertical layer, a sentence stating that the vertical allocations will be described in more detail in section 2.4. Otherwise, an interested/over-eager reader may go searching for that information prematurely (as I just did).

Page 5, Line 16: Is there a citation yet for version 3 of the TNO/MACC inventory?

Page 5, Line 19: It would be nice to have a reminder of the SNAP categories here

Page 5, Line 21: What are the 7 source categories used by the city of Berlin?

[Figure]

Page 5, Line 24: Please go into more detail about how this "mapping" was performed. There may (necessarily) be inherent assumptions here that the reader should know about.

Page 5, Line 28: How was this merging performed? An average of the two at each pixel? Figure 1: In the left image, what do the colors represent? Different source categories or emission fluxes? What are the units? If these really are both in units of flux, then I am wondering why there appear to be more strong point sources in the southern portion of the right image (post-projection) than the left (pre-projection).

Page 6, Lines 8-9: In what context were these flux tower fits performed to tune the remaining VPRM parameters? Was this further part of the SMARTCARB simulations? If so, could we get a little more information about which flux towers were used (or a citation)?

Page 6, Lines 14-15: When the vertical layers were divided up to add new layers, how were the emissions distributed along those layers? For example, if the 4-90m EMEP level was divided into 3 COSMO-GHG levels–4-30, 30-60, and 60-90–were the emissions in 4-30 assigned to be identical, more, or less than those assigned to 60-90?

Page 6, Line 18: Sentence starts with "This modification"; is this referring to the second of the 2 modifications? Or is this intended to read "These modifications"? If it is only referring to the second modification, then where did the 10% number come from in the first modification?

Page 9, Line 14: What is the July difference in ppm, to compare against the 1.6 ppm in the previous (winter) paragraph?

Page 9, Line 32: Is it stated somewhere how the column mean dry air mole fraction is calculated? I assume it is, as the name suggests, the mean concentration at each pixel along the full vertical column for each pixel in the domain, but it is worth stating this explicitly somewhere. Apologies if I just missed it.

Page 10, Lines 19-20: This last sentence is very confusing for me. What does the "95th percentile" refer to?

Page 10, Lines 21-22: Maybe not in this sentence, but somewhere in the next paragraph or two, it would be nice to support the "very large" differences assertion with a number (from Figure 12, presumably).

Page 12, Line 18: Similar to a comment above, I am still not clear on what the "highest percentiles" refers to.

Technical corrections:

Page 1, Line 12: Should read "The results suggest that. . ."

Page 1, Line 18: Comma should be moved, I think: ". . ., the contributions from anthropogenic and natural fluxes and their sensitivity to climate change, and political and societal drivers."

Page 2, Lines 13-16: I suggest putting commas around "such as the WMO's Integrated Global Greenhouse Gas Information System (IG3IS)" to make it an appositive, otherwise the sentence is a run-on.

Page 3, Line 4: "Hogue et al." missing a year

Page 3, Line 8: "Lauvaux et al." also missing a year

Page 4, Line 10: I am not able to make sense of the phrasing "to offload compute intensive part". Is it trying to say "to offload computationally-intensive parts"? If not, is it possible to reword/clarify?

Page 4, Lines 11-12: The phrasing "which allows to increase. . ." feels awkward (and maybe incorrect?). I would consider rephrasing the sentence as "which allows for an increase in spatial and temporal resolution, and which greatly. . ."

Page 4, Line 16: I think it's supposed to be "an NO2" not "a NO2", but I could be wrong.

Page 4, Lines 19-20: "250-km-wide" should be hyphenated to act as one adjective

Page 12, Lines 3, 5, 8, 10, and 12: Might want to say "near" or "around" or some other phrase instead of "of/on the order of" so that readers do not think this is referring to an order of magnitude

Page 12, Line 11: "Bagley et al." does not have a year.

Page 12, Line 25: Change "However, also these methods. . ." to "However, these methods also. . ." to correct awkward phrasing

---

## Referee Comment (RC2) · Anonymous Referee #1 · 30 Dec 2018

General comments. The study aims at understanding the amount of bias in simulating the near-surface CO2 concentrations, related to representation of the elevated anthropogenic CO2 emissions by power plants and industries. Authors report significant differences between the results obtained with common assumption of placing all emissions near surface and with more accurate approach taking into account the stack height and plume rise. The results are useful for sizeable community of CO2 modelers interested in anthropogenic CO2 emissions and their verification with atmospheric measurements, both ground-based and space-based. Authors identify a problem with modeling a plume rise of exhaust by cooling towers, that complicates realistic estimates of CO2 plume injection height. The paper is well written and can be accepted with minor corrections reflecting the comments.

[Figure]

Detailed comments

1.The comparison contrasts sets of CO2 simulations in lowest 20 m near surface made with emissions emitted either at surface or at more realistic heights. It should be mentioned that the observations are often made at higher elevations than 20 m, using either small towers (40-100 m) or tall towers (200-300 m tall). For modeling such observation sites, the conclusions presented in this study can serve more as a warning, rather than ready to use estimate of emission height-related bias.

2.Lagrangian plume models (eg STILT, FLEXPART) are often used in backward, adjoint mode for inverse modeling, and some are used in studies cited here (Page 3 Line 8). In that setting they have to assume emissions are mixed quickly in surface layer of nonzero thickness. It can be as thick as diurnally varying PBL height (Lin et al., 2003) or assigned a constant value (Ganshin et al., 2012). This is done to minimize sampling errors in estimating adjoint tracer concentration near surface, which is made by counting particles in the surface layer. In case of using relatively thick layer, the assumption may reverse the effect of neglecting CO2 emission height, towards having more errors from surface emissions rather than from elevated stacks.

Technical corrections:

Page 2 Line 11 Add period after CO2 and before "Top-down".

Page 3 Line 8 Add year to Lauvaux et al.

Page 3 Line 15 It is worth noting earlier references to air quality modeling, such as SMOKE-CMAQ modeling system (eg Houyoux et al, 2002), to emphasize that the problem had long been recognized and addressed. For CO2 modelling audience it is also useful to mention that in air quality modeling effort is made to account for plume rise height of biomass burning emissions (eg Achtemeier et al, 2010).

Page 4 line 6 Written as "COSMO is the first NWP model worldwide" - it appears that similar effort with ASUCA model (Shimokawabe et al., 2010) was done in about same

time, suggest checking, rephrasing.

Page 7 line 16 Suggest revising "In order to prevent re-heating," as "In order to avoid re-heating,"

Page 12 Line 10 Need to add year to Bagley et al.

References

Achtemeier, G.L.; Goodrick, S.A.; Liu, Y.; Garcia-Menendez, F.; Hu, Y.; Odman, M.T. Modeling Smoke Plume-Rise and Dispersion from Southern United States Prescribed Burns with Daysmoke, Atmosphere, 2011, 2, 358-388.

Ganshin, A., et al: A global coupled Eulerian-Lagrangian model and $1 \times 1$ km CO2 surface flux dataset for high-resolution atmospheric CO2 transport simulations, Geosci. Model Dev., 5, 231-243, https://doi.org/10.5194/gmd-5-231-2012, 2012.

Houyoux, M., Vukovich, J., Seppanen, C., Brandmeyer, J.E., 2002. SMOKE User Manual, MCNC Environmental Modeling Center, 486 pp.

Lin, J. C., C. Gerbig, S. C. Wofsy, A. E. Andrews, B. C. Daube, K. J. Davis, and C. A. Grainger (2003), A near-field tool for simulating the upstream influence of atmospheric observations: The Stochastic Time-Inverted Lagrangian Transport (STILT) model, J. Geophys. Res., 108, 4493, doi:10.1029/2002JD003161, D16.

Shimokawabe T. et al., "An 80-Fold Speedup, 15.0 TFlops Full GPU Acceleration of Non-Hydrostatic Weather Model ASUCA Production Code," SC '10: Proceedings of the 2010 ACM/IEEE International Conference for High Performance Computing, Networking, Storage and Analysis, New Orleans, LA, 2010, pp. 1-11. doi: 10.1109/SC.2010.9

---

## Author Comment (AC1) · 27 Feb 2019

**Anonymous reviewer #2**

We would like to thank the reviewer for the careful reading and constructive comments.

Please find below our point-by-point replies. For clarity, the reviewer's comments are displayed in black, our replies in blue, and suggestions for revised text in blue italics.

**Specific comments:**

Page 4, Line 23: It might be good to include, after the sentence stating the height of the lowest vertical layer, a sentence stating that the vertical allocations will be described in more detail in section 2.4. Otherwise, an interested/over-eager reader may go searching for that information prematurely (as I just did).

We added a corresponding reference to Section 2.4.

Page 5, Line 16: Is there a citation yet for version 3 of the TNO/MACC inventory?

No, Kuenen et al. (2014) referring to version 2 is still the most appropriate reference.

Page 5, Line 19: It would be nice to have a reminder of the SNAP categories here.

We added a reference to Table2 describing the 10 SNAP categories.

Page 5, Line 21: What are the 7 source categories used by the city of Berlin?

We added a reference to the publicly available final report of the SMARTCARB project, where more details on the Berlin inventory (including the mapping of its 7 categories to SNAP) can be found, see Table 2 in the report (https://www.empa.ch/documents/56101/617885/FR_Smartcarb_final_Jan2019.pdf)

Page 5, Line 24: Please go into more detail about how this "mapping" was performed. There may (necessarily) be inherent assumptions here that the reader should know about.

The sentence was not quite correct, since we did not perform a mapping to SNAP but rather a mapping to three broad source groups represented as separate tracers in our simulations. We changed the sentence to the following:

*In our simulations, separate tracers were included for three broad source groups: traffic, industry, heating. The attribution of the 10 SNAP categories in TNO/MACC-3 and the 7 source categories in the Berlin inventory to these groups are described in detail in the final report of the SMARTCARB project (Kuhlmann et al., 2019).*

Page 5, Line 28: How was this merging performed? An average of the two at each pixel? Figure 1: In the left image, what do the colors represent? Different source categories or emission fluxes? What are the units? If these really are both in units of flux, then I am wondering why there appear to be more strong point sources in the southern portion of the right image (post-projection) than the left (pre-projection).

We added the following sentence to clarify the merging:

*The merged emission e per grid cell was estimated as e = (1-f) $e_{TNO}$ + $e_{Berlin}$, where f is the fraction of the grid cell area covered by Berlin.*

Regarding Figure 1: The lefthand figure is only an illustration of the GIS-type of inventory, showing at the same time area, point and line sources, each of them in different units (in kg s$^{-1}$ for point, kg s$^{-1}$ m$^{-1}$ for line and kg s$^{-1}$ m$^{-2}$ for area sources) each being colored separately between minimum and maximum values. In contrast to the right-hand figure, the left-hand figure can thus only be interpreted in a qualitative way.  We will add this information.

Page 6, Lines 8-9: In what context were these flux tower fits performed to tune the remaining VPRM parameters? Was this further part of the SMARTCARB simulations? If so, could we get a little more information about which flux towers were used (or a citation)?

We will add the following sentence explaining the origin of the VPRM parameters used here:

*Model parameter values were taken from a previous study (Kountouris et al., 2018), in which they were optimized using data from 47 European eddy covariance measurement sites for the year 2007.*

Page 6, Lines 14-15: When the vertical layers were divided up to add new layers, how were the emissions distributed along those layers? For example, if the 4-90m EMEP level was divided into 3 COSMO-GHG levels–4-30, 30-60, and 60-90–were the emissions in 4-30 assigned to be identical, more, or less than those assigned to 60-90?

We added the following sentence to further clarify the procedure:
*The attribution of emissions to these sub-layers was done in a rough way, for example by placing a larger proportion of emissions from "combustion in manufacturing industry" into the upper parts but distributing emissions from "Non-industrial (residential) combustion'" rather evenly over the three layers between 4-90m, considering that industrial sources are likely to emit at higher altitudes.*

All information on the finally applied vertical attribution of emissions to the layers is given in Tables 2 and Figure 2 of the manuscript.

Page 6, Line 18: Sentence starts with "This modification"; is this referring to the second of the 2 modifications? Or is this intended to read "These modifications"? If it is only referring to the second modification, then where did the 10% number come from in the first modification?

We changed the wording to "*This latter modification*". The 10% was a very rough assumption to account for $CO_2$ emissions from the waste sector not associated with waste incinerators.

Page 9, Line 14: What is the July difference in ppm, to compare against the 1.6 ppm in the previous (winter) paragraph?

The difference has a strong diurnal cycle in summer. We changed the corresponding sentence as follows:

*In summer, the differences are generally smaller and exhibit a pronounced diurnal cycle. Differences are about 1 ppm at night and almost vanish (about 0.1 ppm) in the afternoon.*

Page 9, Line 32: Is it stated somewhere how the column mean dry air mole fraction is calculated? I assume it is, as the name suggests, the mean concentration at each pixel along the full vertical column for each pixel in the domain, but it is worth stating this explicitly somewhere. Apologies if I just missed it.

The column mean dry air mole fraction is the column integrated amount of $CO_2$ (moles per $m^2$) divided by the column integrated amount of dry air (moles per $m^2$). We will add this information. We changed all references to "column mean" to "column-averaged".

Page 10, Lines 19-20: This last sentence is very confusing for me. What does the "95th percentile" refer to?

We will add "(see last column in Table 4)", since we are referring to the entries in this table here.

Page 10, Lines 21-22: Maybe not in this sentence, but somewhere in the next paragraph or two, it would be nice to support the "very large" differences assertion with a number (from Figure 12, presumably).

Since Figure 12 is only an illustration of the problem, we would like to refrain from providing a number here. The difference is as large as the amplitude of the plume, which, depending on weather situtation, can be up to a few ppm in total column $XCO_2$ and even higher in individual vertical levels (as suggested by Fig. 12).

Page 12, Line 18: Similar to a comment above, I am still not clear on what the "highest percentiles" refers to.

As explained on page 9, we did not only look at mean values but also at different percentiles of the frequency distributions . The statistics for different percentiles are presented in Table 4. We think this should be sufficiently clear.

**Technical corrections:**

Page 1, Line 12: Should read "The results suggest that…"

Done

Page 1, Line 18: Comma should be moved, I think: "…, the contributions from anthropogenic and natural fluxes and their sensitivity to climate change, and political and societal drivers."

Thank you for spotting. Done.

Page 2, Lines 13-16: I suggest putting commas around "such as the WMO's Integrated Global Greenhouse Gas Information System (IG3IS)" to make it an appositive, otherwise the sentence is a run-on.

Commas added.

Page 3, Line 4: "Hogue et al." missing a year, Line 8: "Lauvaux et al." also missing a year

Done

Page 4, Line 10: I am not able to make sense of the phrasing "to offload compute intensive part". Is it trying to say "to offload computationally-intensive parts"? If not, is it possible to reword/clarify?

Yes, that's what it means. "Compute-intensive" is a technically term commonly used in high-performance computing. We added the hyphen and changed "part" to "parts".

Page 4, Lines 11-12: The phrasing "which allows to increase…" feels awkward (and maybe incorrect?). I would consider rephrasing the sentence as "which allows for an increase in spatial and temporal resolution, and which greatly…"

This wording was awkward, indeed. Changed as suggested.

Page 4, Line 16: I think it's supposed to be "an NO2" not "a NO2", but I could be wrong.

You are probably right, changed.

Page 4, Lines 19-20: "250-km-wide" should be hyphenated to act as one adjective

Done

Page 12, Lines 3, 5, 8, 10, and 12: Might want to say "near" or "around" or some other phrase instead of "of/on the order of" so that readers do not think this is referring to an order of magnitude

Thank you. Mostly changed to "around".

Page 12, Line 11: "Bagley et al." does not have a year.

Year added.

Page 12, Line 25: Change "However, also these methods…" to "However, these methods also…" to correct awkward phrasing

Changed as suggested.

---

## Author Comment (AC2) · 27 Feb 2019

**Anonymous reviewer #1**

We would like to thank the reviewer for the careful reading and constructive comments.

Please find below our point-by-point replies. For clarity, the reviewer's comments are displayed in black, our replies in blue, and suggestions for revised text in blue italics.

**Main points**

1.The comparison contrasts sets of CO2 simulations in lowest 20 m near surface made with emissions emitted either at surface or at more realistic heights. It should be mentioned that the observations are often made at higher elevations than 20 m, using either small towers (40-100 m) or tall towers (200-300 m tall). For modeling such observation sites, the conclusions presented in this study can serve more as a warning, rather than ready to use estimate of emission height-related bias.

This is a very valid point. For observations from tall towers the effect will likely be smaller, though not negligible. We added the following sentence at the end of Section 3.3

*The impact on observations from tall tower networks measuring $CO_2$ some 100 m to 300 m above the surface (Bakwin et al., 1995; Andrews et al., 2014) will likely be somewhat smaller than suggested by the numbers above, especially in winter when the atmosphere is less well-mixed.*

and the following sentence in the conclusions section:

*Since measurements of $CO_2$ are often taken from towers some 100 m to 300 m above the surface (Bakwin et al., 1995), the impact on actual ground-based observations will likely be somewhat smaller.*

2.Lagrangian plume models (eg STILT, FLEXPART) are often used in backward, adjoint mode for inverse modeling, and some are used in studies cited here (Page 3 Line 8). In that setting they have to assume emissions are mixed quickly in surface layer of nonzero thickness. It can be as thick as diurnally varying PBL height (Lin et al., 2003) or assigned a constant value (Ganshin et al., 2012). This is done to minimize sampling errors in estimating adjoint tracer concentration near surface, which is made by counting particles in the surface layer. In case of using relatively thick layer, the assumption may reverse the effect of neglecting CO2 emission height, towards having more errors from surface emissions rather than from elevated stacks.

Thank you for pointing this out. In fact, in FLEXPART the particle residence times can be written out for multiple vertical levels, which would offer the possibility to account for emissions at the surface and at higher altitudes separately. To our knowledge, this possibility has not yet been explored in the context of emissions from power plants and industrial sources, though. We added the following sentences:

*In Lagrangian models such as STILT (Lin et al. 2003) or FLEXPART (Stohl et al. 2005), which are often used in backward, adjoint mode for inverse modelling, particles are typically sampled over a fixed vertical depth above the surface or relative to the height of planetary boundary layer to derive source-sensitivities. Similar to the release of emissions at the surface in Eulerian models, this ignores the potentially different sensitivities to emissions from elevated sources.*

**Technical corrections:**

Page 2 Line 11 Add period after CO2 and before "Top-down".

There was already a period.

Page 3 Line 8 Add year to Lauvaux et al.

Year added.

Page 3 Line 15 It is worth noting earlier references to air quality modeling, such as SMOKE-CMAQ modeling system (eg Houyoux et al, 2002), to emphasize that the problem had long been recognized and addressed. For CO2 modelling audience it is also useful to mention that in air quality modeling effort is made to account for plume rise height of biomass burning emissions (eg Achtemeier et al, 2010).

Thank you. We have added these references and modified the paragraph to the following:

*In the air quality modeling community, the importance of vertically distributing emissions has been recognized much earlier (e.g., Houyoux et al. 2002) and is now well established, especially for species such as $SO_2$ that are primarily emitted from power plants and industrial sources (Bieser et al., 2011; Mailler et al., 15 2013; Karamchandani et al., 2014; Guevara et al., 2014). Accounting for plume rise has also been demonstrated to be critical for biomass burning emissions (Achtemeier et al., 2011).*

Page 4 line 6 Written as "COSMO is the first NWP model worldwide" - it appears that similar effort with ASUCA model (Shimokawabe et al., 2010) was done in about same time, suggest checking, rephrasing.

Right, efforts for GPU acceleration of NWP/climate models such as for ASUCA or for the CAM-SE model have indeed been acknowledged in the publication of Fuhrer et al. (2014). Nevertheless, we maintain that COSMO is the first NWP model run on GPUs in an operational weather forecasting context. We slightly changed the wording to " *COSMO is the first operational NWP model worldwide*".

Page 7 line 16 Suggest revising "In order to prevent re-heating," as "In order to avoid re-heating,"

Done

Page 12 Line 10 Need to add year to Bagley et al.

Done